# The CXCL12/CXCR4 Axis in Sepsis-Induced Acute Lung Injury: Mechanisms and Therapeutic Potential

**DOI:** 10.3390/cimb47121052

**Published:** 2025-12-16

**Authors:** Renwei Luo, Qinglu Fan, Qingyun Chen, Zhihao Nie, Lingxuan Dan, Songping Xie

**Affiliations:** Department of Thoracic Surgery, Renmin Hospital of Wuhan University, Wuhan 430000, China; 2023203020029@whu.edu.cn (R.L.);

**Keywords:** CXCL12/CXCR4, acute lung injury/acute respiratory distress syndrome (ALI/ARDS), sepsis, inflammatory, tissue repair

## Abstract

Sepsis remains a major cause of acute lung injury (ALI) and acute respiratory distress syndrome (ARDS), conditions characterized by high mortality and limited therapeutic options. Among the diverse inflammatory pathways implicated in their pathogenesis, the CXCL12/CXCR4 chemokine axis has gained increasing attention for its dual capacity to drive acute inflammation while also supporting tissue repair. Although numerous studies have investigated this signaling pathway, an integrated framework that reconciles its context-dependent functions, upstream regulatory mechanisms, and translational relevance has been lacking. In this review, we synthesize current evidence on the multifaceted roles of the CXCL12/CXCR4 axis in sepsis-induced ALI, highlighting its cell-type-specific effects in neutrophils, macrophages, alveolar epithelial cells, and endothelial cells through downstream pathways such as NF-κB, MAPK, and PI3K/Akt. We further evaluate emerging therapeutic approaches, including small-molecule antagonists (e.g., AMD3100), natural products, and epigenetic modulators. Newly added sections summarize the upstream regulation of CXCL12 by hypoxia, cytokines, and epigenetic factors, discuss the regulatory influence of the alternative receptor CXCR7/ACKR3, and differentiate preclinical insights from human clinical observations. Finally, we outline key obstacles to clinical translation and propose future directions to develop precision medicine strategies that more effectively target this axis. Collectively, our analysis suggests that although the CXCL12/CXCR4 pathway represents a promising target for ALI/ARDS therapy, its context-dependent and cell-specific actions demand carefully tailored modulation rather than uniform inhibition.

## 1. Introduction

Acute lung injury (ALI) and its more severe form, acute respiratory distress syndrome (ARDS), remain major causes of morbidity and mortality in critically ill patients, particularly those with sepsis [1,2]. Despite advances in supportive care, mortality has plateaued, largely because the biological mechanisms underlying sepsis-induced ALI/ARDS are incompletely understood and no effective targeted therapies are currently available [3,4]. The syndrome arises from a convergence of dysregulated inflammation, uncontrolled immune cell recruitment, disruption of the epithelial–endothelial barrier, and impaired tissue repair—all of which are tightly coordinated by chemokine-driven signaling networks [5,6,7,8].

Among these networks, the CXCL12/CXCR4 axis has emerged as a central regulator linking acute inflammation to subsequent repair processes [9]. CXCL12 (stromal cell–derived factor-1, SDF-1) [10] binds the G protein-coupled receptor CXCR4 [11], which is broadly expressed across immune and structural cells in the lung [12,13,14,15]. Beyond directing the trafficking of neutrophils, macrophages, and lymphocytes during injury responses [16,17], CXCL12/CXCR4 signaling modulates cell survival, proliferation, and differentiation via downstream pathways such as NF-κB, MAPK, and PI3K/Akt [18,19].

In the context of sepsis-induced ALI, activation of this axis contributes to inflammatory amplification, endothelial barrier dysfunction, and epithelial injury [20,21,22,23]. Yet paradoxically, growing evidence indicates that CXCL12/CXCR4 also facilitates lung repair by supporting stem cell homing, angiogenesis, and epithelial regeneration [24,25]. Such context-dependent, cell-specific functions—together with complex upstream regulation and interaction with the atypical receptor CXCR7/ACKR3—place this pathway at the interface of lung injury and recovery.

The present review examines the molecular and cellular mechanisms through which the CXCL12/CXCR4 axis shapes the course of sepsis-induced ALI. We summarize the structural and signaling properties of the pathway, synthesize its diverse roles across key pulmonary cell types, and evaluate current therapeutic strategies. In response to recent advances, we also incorporate an updated analysis of upstream regulatory mechanisms and clarify the distinction between preclinical findings and human clinical evidence. By integrating these components, we aim to provide a coherent framework for understanding the therapeutic potential and limitations of targeting CXCL12/CXCR4 in sepsis-associated ALI.

## 2. Structural and Signaling Framework of the CXCL12/CXCR4 Characteristics of CXCL12

### 2.1. Structural Characteristics

Chemokines are a family of small cytokines (8–12 kDa) that guide the migration and positioning of immune and structural cells during inflammation and tissue repair [15,26]. They are classified into four subfamilies—CXC, CC, C, and CX_3_C—based on the arrangement of conserved N-terminal cysteine residues. Members of these families exert their biological functions primarily through binding to their cognate G protein–coupled receptors (GPCRs), which translate extracellular stimuli into intracellular signaling responses [27,28].

CXCL12, or stromal cell-derived factor-1 (SDF-1), is a CXC chemokine consisting of 68 amino acids. Unlike most CXC chemokines encoded on chromosome 4q21, CXCL12 is located on chromosome 10q11, reflecting its evolutionary divergence within the family [29]. Its sole canonical receptor, CXCR4, is a seven-transmembrane GPCR originally identified as a leukocyte-derived receptor (LESTR or Fusin) and later recognized for its role in immune trafficking and development [29,30]. CXCR4 is composed of 352 amino acids and is encoded on chromosome 2q21.

Structural studies have revealed that the interaction between CXCL12 and CXCR4 depends on two key regions of the ligand: the first eight N-terminal amino acids and the RFFESH motif at the C-terminal end of the CXC domain. These domains are required for high-affinity receptor binding and for inducing the conformational changes necessary to initiate CXCR4-dependent intracellular signaling [31,32]. This precise structural complementarity ensures a tightly regulated signaling system that supports both inflammatory and reparative processes in the lung.

### 2.2. Downstream Signaling Pathways

CXCR4 plays a pivotal role in sepsis-induced acute lung injury (SALI) through multiple intracellular signaling pathways, including NF-κB, MAPK, and PI3K/Akt. As a G-protein-coupled receptor (GPCR), CXCR4 activation is initiated by ligand-induced coupling to a heterotrimeric G-protein located on the inner leaflet of the plasma membrane. The G-protein complex, composed of Gα, Gβ, and Gγ subunits, binds Guanosine Diphosphate (GDP)in the resting state. Ligand engagement promotes GDP–Guanosine Triphosphate (GTP) exchange on the Gα subunit, leading to its dissociation from the Gβγ dimer. Based on sequence homology, Gα proteins are classified into four major families: Gαs, Gαi, Gαq, and Gα12.

Upon activation, Gαq stimulates phosphatidylinositol-specific phospholipase C isoforms, predominantly PLCβ, resulting in the hydrolysis of phosphatidylinositol 4,5-bisphosphate (PIP_2_) and the generation of two key second messengers—inositol 1,4,5-trisphosphate (IP_3_) and diacylglycerol (DAG). This pathway elevates intracellular Ca^2+^ levels and activates downstream kinases such as protein kinase C (PKC) [33]. Because intracellular Ca^2+^ flux increases rapidly following chemokine receptor activation, it is widely used as a functional readout of CXCR4 signaling. Activated PKC subsequently engages MAPK cascades, whereas Gαi-coupled receptors activate MAPK through ERK1/2 and Focal Adhesion Kinase (FAK)-dependent mechanisms [34,35,36]. These pathways collectively promote chemotaxis and the production of inflammatory mediators [33,37], thereby facilitating the recruitment of CXCR4-expressing neutrophils, macrophages, and lymphocytes into the inflamed pulmonary parenchyma during sepsis-induced ALI.

Gαq also activates NF-κB through PYK2 [38], which drives inflammasome-related caspase activation (Caspase-1 and Caspase-11) and enhances the release of IL-1β and other cytokines, promoting monocyte recruitment and amplifying lung inflammation. In LPS-induced ALI, NF-κB activation suppresses miR-194 expression in macrophages, resulting in CXCR4 upregulation and augmented inflammatory responses. This shift enhances the production of IL-1β, IL-6, and TNF-α and increases myeloperoxidase activity, further exacerbating lung injury [39].

Downstream of CXCR4, CXCL12 stimulation activates PI3K via both Gβγ and Gα subunits, leading to the phosphorylation of focal adhesion–associated proteins, including Pyk2, FAK, paxillin, Nck, and Crk-L. This pathway promotes CXCR4-dependent chemotaxis and facilitates the recruitment of inflammatory cells into injured lung tissue in sepsis-induced ALI [40]. Concurrently, PI3K-mediated Akt activation—an essential regulator of cell migration, proliferation, and survival—reinforces inflammatory cell accumulation at sites of injury [41]. Notably, CXCL12/CXCR4 signaling also mediates context-dependent cytoprotection, as PI3K/Akt activation inactivates the pro-apoptotic protein Bcl-2 Antagonistic Cell Death (BAD) and upregulates survival-associated genes [42]. Consistent with this dual role, mesenchymal stem cells have been shown to alleviate TNF-α- and LPS-induced apoptosis by activating the CXCL12/CXCR4 axis, thereby reducing lung injury and fibrosis in experimental acute respiratory distress syndrome (ARDS) models [22].

Overall, CXCR4 activation triggers a conserved set of downstream pathways—primarily PI3K/Akt, MAPK/ERK, and NF-κB—that mediate overlapping but context-dependent cellular responses, including chemotaxis, survival, proliferation, and inflammatory gene expression (Figure 1). To avoid redundancy, we summarize these shared signaling nodes here and refer to them in the relevant cell-type sections that follow. In neutrophils, CXCR4-driven PI3K/Akt and MAPK pathways predominantly regulate chemotaxis, degranulation, and NET formation; in macrophages, these pathways intersect with metabolic and epigenetic regulators to influence M1/M2 polarization; in alveolar epithelial cells, PI3K/Akt and Rac1/MMP2 signaling primarily promote migration and survival; and in endothelial cells, NF-κB and PI3K pathways modulate adhesion molecule expression and barrier integrity. Importantly, the qualitative outcome—pro-inflammatory versus reparative—depends on cell type, signal strength, receptor context (CXCR4 vs. CXCR7), and the temporal stage of disease.

### 2.3. Upstream Regulation of CXCL12 in Sepsis and Acute Lung Injury (ALI)

Accumulating evidence indicates that CXCL12 regulation in sepsis-induced acute lung injury (ALI) arises from the convergence of hypoxic, epigenetic, and cytokine-driven signals within the injured lung microenvironment. Tissue hypoxia and microvascular dysfunction—key features of ALI—stabilize Hypoxia-Inducible Factor-1 (HIF-1α), which directly promotes CXCL12 transcription through hypoxia-responsive elements in its promoter, thereby linking local oxygen tension to chemokine accumulation in hypoxic niches [43]. Epigenetic mechanisms provide an additional layer of control: promoter CpG methylation and histone modifications modulate CXCL12 expression in diverse pathological settings, and DNA methylation–mediated transcriptional silencing of CXCL12 has been consistently observed across multiple tissues [44]. Post-transcriptional regulation further fine-tunes CXCL12 levels, with inflammation-responsive microRNAs—beyond the well-characterized miR-194 [39]—including miR-454 [45] contributing to dynamic modulation in epithelial, immune, and stromal compartments.

Recent lung-focused studies also implicate innate immune sensors and the NLRP3 inflammasome in adjusting CXCL12 production during neutrophil-dominant lung injury, highlighting that metabolic stress, epigenetic plasticity, and cell-type-specific inflammatory pathways collectively generate spatially and temporally heterogeneous CXCL12 gradients. These gradients orchestrate immune cell recruitment, vascular responses, and local tissue remodeling in the context of sepsis-induced ALI [21].

## 3. Cell-Type-Specific Roles of the CXCL12/CXCR4 Axis in ALI

In the multifactorial pathogenesis of sepsis-induced acute lung injury (ALI/ARDS), dysfunction across multiple pulmonary cell populations forms the central driver of disease progression. Within this interconnected network, the CXCL12/CXCR4 axis serves as a critical regulatory hub. It governs the recruitment, retention, activation, and reverse transendothelial migration of neutrophils, thereby initiating and amplifying acute inflammatory responses. Simultaneously, it shapes macrophage polarization dynamics, influencing the balance between inflammatory propagation and tissue repair.

Beyond immune cells, CXCL12/CXCR4 signaling also modulates structural cell behavior. In alveolar epithelial cells, it promotes migration and proliferation while suppressing apoptosis, supporting epithelial restitution. In the pulmonary endothelium, the axis regulates inflammatory activation and barrier integrity through coordinated interactions between vascular endothelial cells and infiltrating immune cells.

Through these integrative mechanisms across immune and structural compartments, the CXCL12/CXCR4 axis orchestrates both injury and repair processes in sepsis-induced ALI, underscoring its potential as a compelling therapeutic target.

### 3.1. Acute Lung Injury Neutrophils: Orchestrators of Inflammation and Reverse Migration

Neutrophils, the most abundant innate immune cells in circulation, constitute the frontline effector population in the host response to acute lung injury (ALI) [46]. They are rapidly mobilized to sites of infection or tissue damage, where they mediate pathogen clearance through phagocytosis, the release of proteases and reactive oxygen species (ROS), and the formation of neutrophil extracellular traps (NETs) [47]. While these mechanisms are essential for antimicrobial defense, excessive or prolonged neutrophil activation disrupts the alveolar–capillary barrier, increasing vascular permeability and causing pulmonary edema and impaired gas exchange—ultimately exacerbating inflammation and tissue injury [48].

A growing body of evidence highlights the central role of the CXCL12/CXCR4 axis in regulating neutrophil recruitment, retention, and migratory behavior (Figure 2). In chronic inflammatory lung diseases, CXCR4 expression is markedly elevated on infiltrating neutrophils [49]. Consistently, in both murine and human models of ALI, enhanced neutrophil extravasation into injured lung tissue is accompanied by increased CXCR4 expression, paralleled by elevated CXCL12 levels in pulmonary tissues during the acute inflammatory phase [20]. These observations collectively support the idea that CXCL12/CXCR4 signaling promotes the accumulation and persistent retention of neutrophils within inflamed lung tissue [21,50].

Recent studies further reveal functional heterogeneity within neutrophil populations, with a CXCR4^hi^ subset emerging as a key driver of sepsis-induced ALI [51,52]. Liu et al. reported that LPS stimulation markedly enhances glycolytic activity and NET-forming capacity in CXCR4^hi^ neutrophils, whereas CXCR4^low^ neutrophils exhibit significantly attenuated responses [50]. Complementing these findings, Peng et al. demonstrated that although NLRP3 deficiency does not impair neutrophil migratory ability per se, it substantially reduces pulmonary neutrophil infiltration and ameliorates LPS-induced lung injury by downregulating CXCL12 production and preventing CXCR4 upregulation on neutrophils [21].

Beyond recruitment and retention, CXCR4 also regulates neutrophil reverse transendothelial migration (rTEM). After fulfilling their initial effector functions, neutrophils can re-enter the circulation via rTEM and subsequently redistribute to the lungs, contributing to secondary waves of inflammatory injury [53,54,55]. In an LPS-induced ALI model, Cheng et al. showed that treatment with the CXCR4 antagonist AMD3100 significantly decreased the proportion of rTEM neutrophils in the bloodstream and mitigated histopathological lung damage [56]. These findings underscore the essential role of CXCR4 in governing neutrophil trafficking dynamics and amplifying lung inflammation.

In summary, the CXCL12/CXCR4 signaling axis orchestrates the recruitment, retention, activation, and reverse migration of neutrophils, thereby shaping both the onset and progression of sepsis-induced ALI (Figure 2). These mechanistic insights position the CXCL12/CXCR4 axis as a compelling therapeutic target for interventions aimed at modulating neutrophil-driven lung injury.

### 3.2. Macrophages: Balancing M1/M2 Polarization

Macrophages occupy a central immunomodulatory position in the pathogenesis of acute lung injury (ALI). By sensing pathogen-associated molecular patterns (PAMPs), they initiate innate immune signaling cascades and release a broad array of inflammatory mediators [57,58]. In sepsis-induced acute lung injury (SALI), macrophage polarization profoundly shapes the course of inflammation. Hyperactivation of pro-inflammatory M1 macrophages—driven largely by LPS/NF-κB signaling—results in excessive production of IL-1β and TNF-α, disruption of the alveolar–vascular barrier, pulmonary edema, and impaired oxygenation [59,60]. Conversely, delayed or insufficient activation of anti-inflammatory and reparative M2 macrophages—regulated by IL-4/PPARγ signaling—limits the resolution of inflammation and compromises tissue repair [61,62].

Macrophage polarization is co-regulated by metabolic and epigenetic mechanisms. M1 macrophages predominantly rely on glycolysis, whereas M2 macrophages depend on oxidative phosphorylation [63]. Epigenetic regulators, including microRNAs [64,65,66] and m^6A RNA methylation [67], exert additional control over macrophage phenotype and function. M2 macrophages further contribute to tissue repair by secreting mediators such as CXCL12, IL-1α, PDGF, and TGF-β1, which suppress inflammation and promote angiogenesis and tissue remodeling [68].

Emerging studies implicate the CXCL12/CXCR4 axis in governing macrophage polarization during ALI (Figure 2). In injured lung tissues, CXCR4 expression is markedly elevated on macrophages; its knockdown reduces the production of IL-6 and TNF-α while increasing IL-10 expression, shifting the balance toward an anti-inflammatory phenotype [69]. Moreover, miR-194 has been shown to attenuate macrophage-mediated inflammation by suppressing CXCR4 expression [39]. Collectively, these findings suggest that CXCL12/CXCR4 signaling influences macrophage polarization, potentially through epigenetic mechanisms, thereby contributing to the development and progression of SALI [39,69].

### 3.3. Alveolar Epithelial Cells: Guardians of Barrier and Repair

Alveolar epithelial cells (AECs) are essential for maintaining alveolar structure and gas exchange and are broadly classified into type I (AT I) and type II (AT II) cells. AT I cells form the air–blood barrier and are indispensable for efficient oxygen diffusion, whereas AT II cells function as progenitors for AT I cells, producing pulmonary surfactant and initiating proliferation and differentiation after injury to restore epithelial integrity [70]. In sepsis-induced ALI/ARDS, AECs—particularly the highly vulnerable AT I population—sustain extensive injury and detachment, exposing the basement membrane and leading to increased permeability, alveolar flooding, and refractory hypoxemia [3,71]. The degree of epithelial disruption is closely correlated with disease severity, and timely epithelial repair is essential for edema resolution and recovery [72].

Accumulating evidence indicates that the CXCL12/CXCR4 axis plays an important role in epithelial repair after lung injury (Figure 3). CXCL12/CXCR4 signaling enhances epithelial migration and regeneration through activation of Rac1 and matrix metalloproteinase-2 (MMP-2), while hypoxia-inducible factor-1 (HIF-1) upregulates CXCL12 and CXCR4 expression in AT II cells, promoting cell spreading and reducing alveolar permeability and edema formation [9,43]. In addition, this axis confers cytoprotective effects by inhibiting epithelial apoptosis [21,22]; for example, mesenchymal stem cells (MSCs) exert protective actions on alveolar epithelium partly through activation of CXCL12/CXCR4 signaling in ARDS models [43]. Notably, some studies suggest that CXCL12/CXCR4 functions in a context-dependent manner; in certain postoperative lung injury models, the pathway primarily enhances epithelial migration rather than proliferation [73].

In summary, CXCL12/CXCR4 signaling regulates AT II cell migration, proliferation, and survival, thereby playing a pivotal role in alveolar epithelial repair during sepsis-induced ALI (Figure 3). These findings highlight the axis as a promising therapeutic target for interventions aiming to promote epithelial restitution and improve clinical outcomes.

### 3.4. Pulmonary Vascular Endothelial Cells: A Nexus of Barrier Dysfunction and Potential Regeneration

Another key pathological feature of ALI/ARDS is the disruption of pulmonary endothelial barrier integrity [3,74]. In sepsis-induced ALI/ARDS, pathogen-associated molecules such as endotoxins activate NF-κB and MAPK signaling pathways through TLR4, driving endothelial cells toward pro-inflammatory and pro-coagulant phenotypes. This activation markedly upregulates adhesion molecules, including intercellular adhesion molecule-1 (ICAM-1), vascular cell adhesion molecule-1 (VCAM-1), and E-selectin, thereby enhancing neutrophil adhesion to the vascular endothelium [75,76,77]. Concurrently, inflammatory mediators and enzymatic degradation contribute to glycocalyx shedding, loss of tight junction proteins, and cytoskeletal contraction—events that collectively increase vascular permeability and lead to non-cardiogenic pulmonary edema [78,79]. In addition, endothelial cells downregulate anticoagulant mediators and upregulate procoagulant factors, promoting microthrombus formation and impairing microcirculatory perfusion [80,81]. Ultimately, the reciprocal interactions among endothelial cells, immune cells, and stromal cells via inflammatory cytokines and reactive oxygen species establish a self-perpetuating cycle of injury that further compromises the alveolar–capillary barrier and may contribute to subsequent fibrotic remodeling [76].

The CXCL12/CXCR4 signaling pathway may play multiple roles in regulating endothelial function (Figure 4). Evidence from models such as cerebral ischemia suggests that this pathway supports angiogenesis and contributes to tissue repair [76,77,82]. During inflammatory responses, METTL14-mediated m^6^A methylation enhances CXCR4 mRNA stability, thereby amplifying endothelial inflammation [83]. Moreover, endothelial-derived CXCL12 can recruit NK cells to sites of damage, whereas CXCR4 deficiency attenuates this protective response [84]. These findings collectively indicate that the CXCL12/CXCR4 axis may influence endothelial biology through diverse mechanisms, including immune cell chemotaxis, angiogenic regulation, and modulation of inflammatory signaling.

Despite these insights, the specific regulatory roles of CXCL12/CXCR4 signaling in pulmonary vascular endothelial cells during sepsis-induced ALI remain incompletely defined. Key unresolved questions include whether this pathway affects pulmonary microvascular angiogenesis, shapes immune–endothelial interactions, or modulates structural components such as glycocalyx integrity and cytoskeletal stability. Addressing these knowledge gaps may provide mechanistic insight into ALI/ARDS pathogenesis and inform the development of endothelial-targeted therapeutic strategies.

In summary, although the CXCL12/CXCR4 axis is implicated in endothelial dysfunction in sepsis-induced ALI, its mechanistic contributions are less well characterized compared with its roles in immune and epithelial cell regulation. Future studies should clarify: (1) whether CXCL12/CXCR4 signaling directly regulates endothelial glycocalyx maintenance and junctional protein stability; (2) how endothelial-derived CXCL12 shapes communication with circulating and resident immune cells to propagate or resolve injury; and (3) whether this axis promotes maladaptive angiogenesis or vascular regeneration during ALI recovery. Elucidating these mechanisms will fill an important knowledge gap and may identify promising endothelial-directed therapeutic targets.

### 3.5. CXCR7/ACKR3: Modulator of CXCL12 Bioavailability and Signaling Bias

CXCR7 (ACKR3) functions as an atypical chemokine receptor with high affinity for CXCL12, acting predominantly as a scavenger and signaling modulator rather than a classical G-protein–coupled chemoattractant receptor [85,86]. Inflammatory disorders—including inflammatory bowel disease, rheumatoid arthritis, and various forms of encephalitis—demonstrate CXCR7 expression across multiple cell types, most notably endothelial cells, selected immune cell subsets, and specific stromal populations [87,88,89]. However, quantitative and cell-type–resolved data in the context of human sepsis or acute lung injury (ALI) remain limited and at times inconsistent. Mechanistically, CXCR7 modulates CXCL12 biology through several pathways: (i) reducing extracellular CXCL12 availability via ligand sequestration and internalization; (ii) forming heterodimers with CXCR4, thereby shifting downstream signaling toward β-arrestin–mediated pathways rather than canonical Gαi coupling [90]; and (iii) responding to biased agonists that preferentially activate β-arrestin–associated programs linked to tissue protection and repair [91].

Despite these insights, several important knowledge gaps persist. First, comprehensive, cell-type–specific quantification of CXCR7 expression in human and murine lungs during sepsis-induced ALI is lacking. Future studies integrating single-cell RNA sequencing with targeted immunostaining will be critical for defining its spatial and temporal distribution. Second, although co-immunoprecipitation and resonance energy transfer assays support CXCR7–CXCR4 heterodimerization in heterologous systems, functional validation of these complexes in primary pulmonary cell types under ALI conditions remains sparse. Third, while CXCR7-biased agonists—and to some degree antagonists—have shown preclinical potential in promoting reparative signaling, studies specifically evaluating their effects in sepsis-induced ALI models are insufficient [92].

Taken together, these considerations highlight the need for future therapeutic strategies targeting the CXCL12 axis to incorporate both CXCR4 and CXCR7. Systematic characterization of CXCR7 expression, functional interrogation of CXCR4–CXCR7 interactions in pulmonary cells, and evaluation of biased agonists in clinically relevant models of lung injury and repair will be essential for advancing this field.

## 4. Clinical Translation Perspective: From Mechanistic Insights to Therapeutic Strategies

### 4.1. Distinguishing Preclinical Evidence from Human Data; Circulating CXCL12/CXCR4 in Human Sepsis/Acute Lung Injury (ALI)

Most mechanistic insights and therapeutic evaluations of the CXCL12/CXCR4 axis in ALI originate from murine lipopolysaccharide (LPS) and cecal ligation and puncture (CLP) models, as well as from in vitro studies using primary cells or established cell lines [20,93]. Although these models are indispensable for dissecting signaling pathways and cellular interactions, they possess inherent limitations. Species-specific differences in immune composition, neutrophil lifespan and metabolic properties, the magnitude of injury, and lung structural organization all constrain direct translational relevance to human disease [94,95].

By contrast, human data remain limited. Small observational studies have reported altered circulating CXCL12 levels in sepsis and in subgroups of patients with acute lung injury or acute respiratory distress syndrome (ARDS) [96], but these findings are inconsistent—likely reflecting variability in sampling time points (acute versus recovery phase), infection source, patient heterogeneity, and assay methodologies. Only a few studies have quantified CXCL12 or CXCR4 expression directly in human lung tissues or characterized temporal changes during disease progression, and interventional clinical data targeting this axis in sepsis or ALI are effectively nonexistent.

Accordingly, in this review, we clearly distinguish between preclinical and human/clinical evidence to avoid overinterpretation of animal-based findings. Future research should prioritize large-scale, longitudinal patient studies integrating plasma biomarker profiling with bronchoalveolar sampling and single-cell tissue analyses. Such efforts will be essential to validate mechanisms inferred from animal models and to identify clinically actionable biomarkers capable of guiding the timing and selection of CXCL12/CXCR4-targeted therapies.

### 4.2. Therapeutic Modulation of the CXCL12/CXCR4 Axis: Balancing Inflammation Control and Tissue Repair

In the complex pathophysiology of acute lung injury/acute respiratory distress syndrome (ALI/ARDS), the CXCL12/CXCR4 signaling axis exerts context-dependent effects during both injury and repair, making it a compelling yet challenging therapeutic target. Current approaches to modulating this axis are diverse. For example, the protein tyrosine phosphatase PTP1B attenuates CXCR4-mediated signaling and promotes neutrophil senescence [93]. Natural compounds such as the Epimedium-derived flavonoid Icariside II (ICS II) selectively inhibit neutrophil CXCR4, thereby reducing neutrophil extracellular trap (NET) formation [97]. The anti-allergic drug Tranilast suppresses inflammation by inhibiting the CXCR4/JAK2/STAT3 signaling cascade [23]. In addition, miR-454 overexpression and Resolvin D1 (RvD1) have been shown to mitigate lung inflammation via modulation of this axis [45,98]. Conversely, during the reparative phase, tissue factor pathway inhibitor enhances mesenchymal stem cell (MSC) migration, homing, and proliferation by activating CXCL12/CXCR4 signaling, thereby facilitating tissue regeneration.

Compared with these investigational modulators—which each fine-tune specific components of the axis—the clinically approved CXCR4 antagonist AMD3100 (Plerixafor) employs a fundamentally different mechanism. AMD3100 acts as a potent, direct blocker of CXCR4, producing a broad and immediate disruption of CXCL12 signaling. Its therapeutic value lies in its ability to reprogram immune cell trafficking [99,100]. Beyond limiting the recruitment of inflammatory cells into the lung, AMD3100 uniquely promotes the reverse transendothelial migration (rTEM) of infiltrated neutrophils and monocytes, effectively clearing these cells from the tissue and dismantling pro-inflammatory niches within the lung microenvironment [54].

This mechanism stands in sharp contrast to classical anti-inflammatory therapies—such as corticosteroids or anti-IL-6/TNF-α antibodies—which primarily suppress cytokine signaling in a global manner. Although such “signal-dampening” strategies can reduce inflammation, they do not correct the mislocalization of inflammatory cells and may inadvertently suppress essential reparative programs, thereby increasing susceptibility to infection [101]. In contrast, AMD3100 targets the spatial and migratory behavior of immune cells, aiming to evacuate pathogenic cells from injured tissues and restore the conditions necessary for repair.

However, the potency of AMD3100 also poses challenges. Because the CXCL12/CXCR4 axis is essential for reparative processes such as stem cell homing and progenitor cell recruitment [102], indiscriminate or prolonged blockade may impair recovery. In this context, agents such as PTP1B inhibitors or ICS II may offer more nuanced modulation with fewer risks to repair mechanisms. Thus, the therapeutic benefit of AMD3100 depends critically on timing; it is best suited for the acute inflammatory phase, where targeted disruption of CXCL12/CXCR4-driven immune cell retention is advantageous.

Collectively, the dual functions of the CXCL12/CXCR4 axis underscore the need for precise and context-specific therapeutic strategies. Simple global inhibition or activation is unlikely to be universally beneficial because the axis simultaneously contributes to both lung injury and tissue repair [29]. A comprehensive understanding of the mechanistic differences among CXCL12/CXCR4 modulators—including classical antagonists, pathway-specific inhibitors, and bioactive natural compounds—is therefore essential for designing therapies aligned with distinct stages of ALI/ARDS pathogenesis. In the following section, we integrate evidence across these therapeutic classes and present a comparative framework (Table 1) to clarify their respective positions within the evolving drug-development landscape.

## 5. Conclusions

The CXCL12/CXCR4 axis functions as a central regulatory hub in sepsis-induced acute lung injury (ALI), orchestrating the balance between inflammatory injury and tissue repair. Its context-dependent duality presents both a promising therapeutic opportunity and a significant challenge: how to selectively suppress its detrimental pro-inflammatory signaling while preserving—or even enhancing—its reparative functions. A range of therapeutic approaches, spanning small-molecule antagonists such as AMD3100, bioactive natural compounds, and epigenetic regulators, have demonstrated substantial efficacy in preclinical models. However, successful clinical translation will require several key barriers to be addressed.

Future work should prioritize the development of cell-specific delivery strategies—for example, nanoparticle-based systems conjugated with lung-homing peptides—to achieve spatial and cellular precision in modulating this axis. Equally important is the definition of an optimal therapeutic window: inhibiting CXCL12/CXCR4 signaling during the acute hyperinflammatory phase while potentially restoring or amplifying its activity during the reparative phase. Advances in single-cell multi-omics, spatial transcriptomics, and integrative bioinformatics will be essential for delineating the cell-type- and stage-specific signaling networks governed by this pathway.

Ultimately, the adoption of precision medicine strategies—guided by biomarkers that reflect the dominant functional role of the CXCL12/CXCR4 axis in individual patients—may be critical for realizing the full therapeutic potential of targeting this pathway in ALI/ARDS.

## Figures and Tables

**Figure 1 cimb-47-01052-f001:**
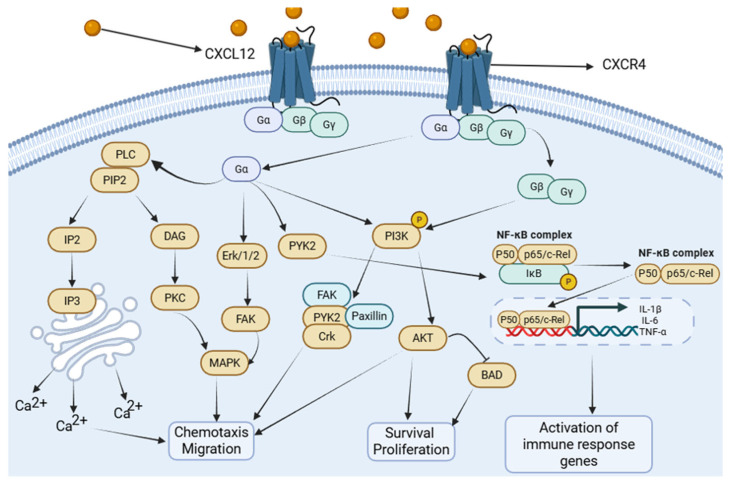
CXCL12/CXCR4 signaling cascade. CXCL12 binding to CXCR4 triggers the dissociation of heterotrimeric G-proteins (Gαi, Gβγ). This activates major downstream cascades: (1) The PI3K/Akt pathway, promoting cell survival, migration, and metabolic changes. (2) The MAPK/ERK pathway, driving proliferation, cytokine production, and chemotaxis. (3) The NF-κB pathway, leading to the transcription of pro-inflammatory mediators. (4) PLC activation, resulting in calcium flux and PKC activation. These pathways collectively regulate immune cell recruitment, inflammation, and tissue repair in ALI. Created in Biorender. Renwei Luo. (2025) https://biorender.com. Arrows indicate the direction of cellular migration or signaling events. Solid arrows denote activation or promotion of biological processes, whereas dashed arrows represent indirect or putative interactions. Labels connected by arrows identify the structure being indicated.

**Figure 2 cimb-47-01052-f002:**
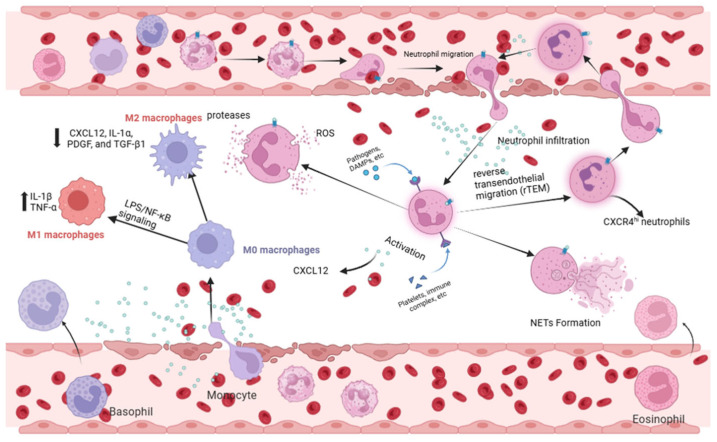
Neutrophil and macrophage responses mediated by CXCL12/CXCR4 in ALI. (Right) In neutrophils, CXCL12/CXCR4 signaling drives chemotaxis, tissue infiltration, and activation (NETosis, ROS production). It also regulates the reverse transendothelial migration (rTEM) of neutrophils, contributing to systemic inflammation. (Left) In macrophages, this axis influences polarization. It can potentiate M1 pro-inflammatory polarization (via NF-κB), enhancing cytokine storm. Conversely, it may support M2 anti-inflammatory/reparative polarization in certain contexts, facilitating tissue repair. Created in Biorender. Renwei Luo. (2025) https://biorender.com. Arrows indicate the direction of cellular migration or signaling events. Solid arrows denote activation or promotion of biological processes, whereas dashed arrows represent indirect or putative interactions. Labels connected by arrows identify the structure being indicated.

**Figure 3 cimb-47-01052-f003:**
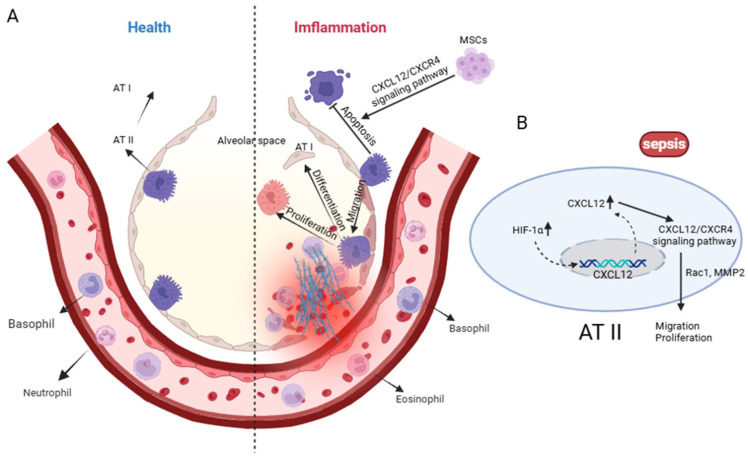
Alveolar Epithelial Cells: Guardians of Barrier and Repair. (**A**) Under physiological conditions, alveolar type I (AT I) and type II (AT II) epithelial cells maintain alveolar structural integrity with minimal immune-cell infiltration. During sepsis, excessive activation of the CXCL12/CXCR4 axis promotes AT II cell proliferation, migration and differentiation toward AT I, while inducing apoptosis of AT I cells and recruiting large numbers of granulocytes. Mesenchymal stem cells (MSCs) can mitigate epithelial injury by suppressing AT II apoptosis through CXCL12/CXCR4-dependent signaling. (**B**) Sepsis-related hypoxia upregulates Hypoxia-Inducible Factor-1α (HIF-1α) and CXCL12, and CXCR4 activation triggers Rac1/MMP2-dependent cytoskeletal remodeling to support AT II-driven alveolar repair. Created in Biorender. Renwei Luo. (2025) https://biorender.com. Hypoxia-Inducible Factor-1(HIF-1); Ras-related C3 botulinum toxin substrate 1 (Rac1); Matrix Metallopeptidase 2 (MMP2). Arrows indicate the direction of cellular migration or signaling events. Solid arrows denote activation or promotion of biological processes, whereas dashed arrows represent indirect or putative interactions. Labels connected by arrows identify the structure being indicated.

**Figure 4 cimb-47-01052-f004:**
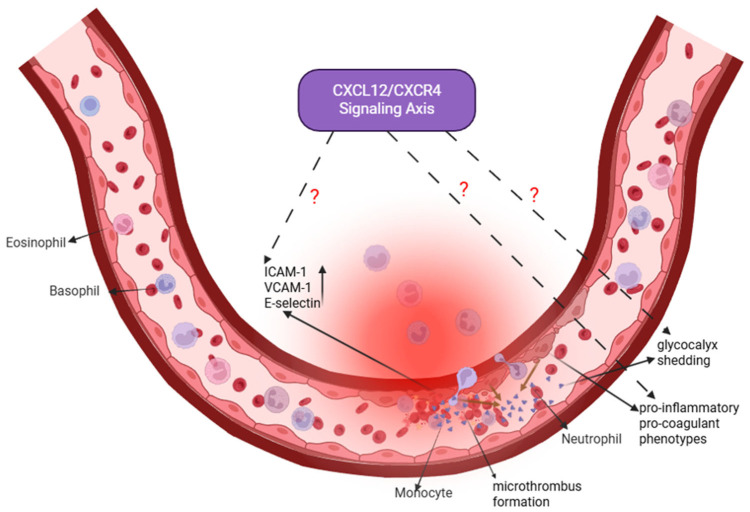
Pulmonary Vascular Endothelial Cells: A Nexus of Barrier Dysfunction and Potential Regeneration. In sepsis-induced ALI, pulmonary endothelial cells undergo activation, leading to increased expression of adhesion molecules (intercellular adhesion molecule-1 (ICAM-1), vascular cell adhesion molecule-1 (VCAM-1), E-selectin), glycocalyx shedding, a shift toward a pro-inflammatory and pro-coagulant phenotype, and microthrombus formation. These changes collectively disrupt barrier integrity and promote edema. The CXCL12/CXCR4 signaling axis (highlighted in purple) is implicated in endothelial dysfunction, but its precise mechanistic contributions remain to be fully elucidated. Dashed lines with question marks indicate specific areas of uncertainty, including how this axis regulates glycocalyx/junctional integrity, communicates with immune cells, and influences vascular outcomes (pathological angiogenesis versus regeneration). Addressing these gaps is crucial for developing endothelial-targeted therapies. Created in Biorender. Renwei Luo (2025) https://biorender.com. Arrows indicate the direction of cellular migration or signaling events. Solid arrows denote activation or promotion of biological processes, whereas dashed arrows represent indirect or putative interactions. Labels connected by arrows identify the structure being indicated. The dark brownish-tan arrow represents the process of endothelial glycocalyx shedding.

**Table 1 cimb-47-01052-t001:** Therapeutic Strategies Targeting the CXCL12/CXCR4 Axis and Related Pathways in ALI/ARDS.

Drug Class/Strategy (Examples)	Mechanistic Focus in ALI/ARDS	CXCL12/CXCR4 Modulation	Key Evidence	Major Limitations	References
Direct CXCR4 Antagonists (AMD3100)	Blocks CXCL12–CXCR4; reduces leukocyte recruitment; promotes neutrophil reverse migration	Direct, complete inhibition	Reduces inflammation and fibrosis in LPS/CLP/viral ALI; clinically approved for mobilization	May hinder repair (stem cell homing); timing-sensitive; potential immune disruption	[54,56,99,100,103]
Natural Compounds/SPMs (ICS II, RvD1)	ICS II inhibits NETs; RvD1 enhances resolution pathways	Indirect/partial modulation	Reduce neutrophil activation and lung injury in rodent ALI	Multi-target actions; variable purity and bioavailability	[97,98]
Epigenetic/Molecular Modulators (miR-194/454, PTP1B inhibitors)	Reprogram immune inflammation; restrain CXCR4 signaling; neutrophil senescence	Upstream tuning of CXCR4 expression/signaling	Improve permeability, cytokine burden, and survival	Delivery limitations; off-target effects; early translational stage	[39,45,94]
Corticosteroids (Dexamethasone)	Broad suppression of NF-κB/MAPK and cytokines	Indirect suppression	Proven benefit in severe ARDS (COVID-19 & non-COVID trials)	Infection risk; hyperglycemia; non-specific; may delay repair	[104,105,106]
Other Targeted Agents (CXCR1/2 inhibitors; Statins)	Block CXCL8–driven neutrophil trafficking; stabilize endothelium	Pathway-adjacent effects intersecting CXCL12/CXCR4	CXCR1/2 inhibitors effective in murine ALI; statins mixed/negative in ARDS	Limited efficacy/safety for CXCR1/2i; statins lack mortality benefit	[107,108,109,110,111]

Abbreviations: ALI, acute lung injury; ARDS, acute respiratory distress syndrome; LPS, lipopolysaccharide; CLP, cecal ligation and puncture; NETs, neutrophil extracellular traps; SPMs, specialized pro-resolving mediators; ICS II, Icariside II; RvD1, Resolvin D1; NF-κB, nuclear factor kappa-light-chain-enhancer of activated B cells; MAPK, mitogen-activated protein kinase; CXCL8, C-X-C motif chemokine ligand 8; CXCR1/2, C-X-C chemokine receptor type 1/2.

## Data Availability

No new datasets were created or analyzed in this study. All materials supporting the research findings are included within the references of this article.

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
