# Peer review of "The CXCL12/CXCR4 Axis in Sepsis-Induced Acute Lung Injury: Mechanisms and Therapeutic Potential"

_cimb, 2025, doi:10.3390/cimb47121052_

Round 1

Reviewer 1 Report

Comments and Suggestions for Authors

Hello,

 Congratulations on a brief review of an important immunomodulator, AMD3100, and its potential applications in combating sepsis-induced ALI/ARDS. I will definitely review the topic in my cellular physiology/Immunology lectures in graduate and medical school.

The write-up flows nicely, although some areas require polish:

Lines 110-125 – relevant but hanging information in short/broken sentences.

Lines 212-213 – Redundant abbreviations

Lines 159 and several others – Citing the original references will help the readers with more details.

Comments on the Quality of English Language

Please refer to my recommendation to authors. 

Author Response

1) “Lines 110–125 – relevant but hanging information in short/broken sentences.”

We thank the reviewer for this helpful suggestion. The sentences in Lines 110–125 have been revised to improve coherence and readability. Specifically, fragmented statements were consolidated into complete sentences with clearer logical transitions, while preserving the original scientific content.

Manuscript changes: 118-136(location of the modified content)

Activation of the NF-κB pathway in LPS-induced acute lung injury (ALI) has been shown to suppress miR-194 expression, resulting in upregulation of CXCR4 in macrophages and subsequent amplification of pro-inflammatory signaling. This shift enhances the production of IL-1β, IL-6, and TNF-α and increases myeloperoxidase activity, thereby aggravating pulmonary inflammation and tissue injury. Downstream of CXCR4, CXCL12 stimulation activates phosphoinositide 3-kinase (PI3K) via both Gβγ and Gα subunits, leading to phosphorylation of focal adhesion–related proteins, including Pyk2, focal adhesion kinase (FAK), paxillin, Nck, and Crk-L.

This signaling cascade promotes CXCR4-dependent chemotaxis of inflammatory cells and facilitates their recruitment to injured lung tissue in sepsis-induced ALI. In parallel, PI3K-driven activation of AKT—a key regulator of cell migration, proliferation, and survival—further reinforces inflammatory cell accumulation at sites of injury. Notably, CXCL12/CXCR4 signaling also mediates context-dependent cytoprotective effects, as PI3K/AKT activation can inactivate the pro-apoptotic protein BAD and induce the expression of cell survival–associated genes. Consistent with this dual functionality, mesenchymal stem cells have been reported to attenuate TNF-α– and LPS-induced apoptosis through activation of the CXCL12/CXCR4 axis, thereby alleviating lung injury and pulmonary fibrosis in experimental ARDS models [43].

2) “Lines 212–213 – Redundant abbreviations.”

We agree with the reviewer. Redundant abbreviations in Lines 212–213 have been removed, and abbreviations are now defined only once at their first appearance for clarity and consistency.

Manuscript changes: 269-270(location of the modified content)

In sepsis-induced ALI/ARDS, alveolar epithelial cells—particularly AT I cells—undergo severe injury and detachment, leading to basement membrane exposure, increased permeability, and alveolar edema, ultimately resulting in refractory hypoxemia

3) “Lines 159 and several others – Citing the original references will help the readers with more details.”

We appreciate this suggestion. We have revised Line 159 and several related sections by adding citations to the original research articles, in addition to review references, to facilitate readers’ access to primary experimental evidence.

Manuscript changes: 207-208(location of the modified content)

These findings collectively suggest that the CXCL12/CXCR4 axis promotes neutrophil accumulation and retention in inflammatory lung injury [21,51].

Reviewer 2 Report

Comments and Suggestions for Authors

The authors reviewed the role of the CXCL12/CXCR4 axis in sepsis-induced acute lung injury (ALI). The topic is clinically important; however, the current version requires substantial revisions before it can be considered suitable for publication.

Although the authors have reviewed lots of references, the manuscript largely reads as a descriptive narrative rather than a critical, integrative review. I have the following major concerns:

  1. All the figures have only one brief title. They require brief figure legends to highlight the key points for these figures. The authors are also required to show which software they used to prepare these figures.
  2. Figures 3 and 4 do not visually highlight the role of CXCL12/CXCR4 signaling, instead depicting general ALI biology without mechanistic specificity.
  3. Table 1 has not been well prepared, unbalanced. It should highlight the mechanism more, not the reference.
  4. The review repeatedly restates the same pathways (NF-κB, MAPK, PI3K/Akt) across multiple sections, reducing clarity and cohesion.
  5. This manuscript mentioned that CXCR7/ACKR3 modulates CXCL12 availability, scavenging, and signaling bias, but the authors may need more data about (1) CXCR7 expression in lung cells; (2) CXCR7–CXCR4 heterodimerization; (3) CXCR7-biased agonists.

  6. Most cited studies are animal-based. The review needs to clearly distinguish preclinical findings from human observational or interventional data. The authors had better add a short section on circulating CXCL12/CXCR4 levels in human sepsis/ALI, and then discuss strengths/limitations of those preclinical models.
  7. Discussion on Upstream Regulation of CXCL12 in Sepsis/ALI is missing. The authors had better mention (1) HIF-1α induction by hypoxia (important given ALI hypoxemia); (2) Epigenetic regulation (e.g., methylation, microRNAs beyond miR-194); (3) Cytokine-driven CXCL12 transcription; (4) A more complete view of CXCL12 regulation is needed

Comments on the Quality of English Language

Professional English editing is needed.

Author Response

1) “All the figures have only one brief title. They require brief figure legends to highlight the key points for these figures. The authors are also required to show which software they used to prepare these figures.”We thank the reviewer for this suggestion. We have added concise, informative legends for all figures emphasizing the mechanistic role of CXCL12/CXCR4 where relevant, and included a statement in Methods acknowledging the software used to prepare and assemble the figures.

Manuscript changes — Figure legends (replace current single-line titles with these legends; insert into manuscript right after each figure):

Figure 1. CXCL12/CXCR4 signaling cascade.
Legend: CXCL12 binding to CXCR4 triggers the dissociation of heterotrimeric G-proteins (Gαi, Gβγ). This activates major downstream cascades: 1) The PI3K/Akt pathway, promoting cell survival, migration, and metabolic changes. 2) The MAPK/ERK pathway, driving proliferation, cytokine production, and chemotaxis. 3) The NF-κB pathway, leading to the transcription of pro-inflammatory mediators. 4) PLC activation, resulting in calcium flux and PKC activation. These pathways collectively regulate immune cell recruitment, inflammation, and tissue repair in ALI. Figure created with BioRender.com.

Figure 2. Neutrophil and macrophage responses mediated by CXCL12/CXCR4 in ALI.
Legend: (Left) In neutrophils, CXCL12/CXCR4 signaling drives chemotaxis, tissue infiltration, and activation (NETosis, ROS production). It also regulates the reverse transendothelial migration (rTEM) of neutrophils, contributing to systemic inflammation. (Right) In macrophages, this axis influences polarization. It can potentiate M1 pro-inflammatory polarization (via NF-κB), enhancing cytokine storm. Conversely, it may support M2 anti-inflammatory/reparative polarization in certain contexts, facilitating tissue repair. Figure created with BioRender.com.

Figure 3. Alveolar Epithelial Cells: Guardians of Barrier and Repair.
Legend: (A) Under physiological conditions, alveolar type I (AT I) and type II (AT II) epithelial cells maintain alveolar structural integrity with minimal immune-cell infiltration. During sepsis, excessive activation of the CXCL12/CXCR4 axis promotes AT II cell proliferation, migration and differentiation toward AT I, while inducing apoptosis of AT I cells and recruiting large numbers of granulocytes. Mesenchymal stem cells (MSCs) can mitigate epithelial injury by suppressing AT II apoptosis through CXCL12/CXCR4-dependent signaling. (B) Sepsis-related hypoxia upregulates HIF-1α and CXCL12, and CXCR4 activation triggers Rac1/MMP2-dependent cytoskeletal remodeling to support AT II-driven alveolar repair.  Figure created with BioRender.com.

Figure 4. Pulmonary Vascular Endothelial Cells: A Nexus of Barrier Dysfunction and Potential Regeneration.

Legend: In sepsis-induced ALI, pulmonary endothelial cells undergo activation, leading to increased expression of adhesion molecules (ICAM-1, VCAM-1, E-selectin), glycocalyx shedding, a shift toward a pro-inflammatory and pro-coagulant phenotype, and microthrombus formation. These changes collectively disrupt barrier integrity and promote edema. The CXCL12/CXCR4 signaling axis (highlighted in purple) is implicated in endothelial dysfunction, but its precise mechanistic contributions remain to be fully elucidated. Dashed lines with question marks indicate specific areas of uncertainty, including how this axis regulates glycocalyx/junctional integrity, communicates with immune cells, and influences vascular outcomes (pathological angiogenesis versus regeneration). Addressing these gaps is crucial for developing endothelial-targeted therapies. Figure created with BioRender.com.

2) “Figures 3 and 4 do not visually highlight the role of CXCL12/CXCR4 signaling, instead depicting general ALI biology without mechanistic specificity.”We appreciate this point. We have revised Figures 3 and 4 to explicitly map CXCL12/CXCR4 signaling onto epithelial and endothelial mechanisms respectively. The legends and figure content now emphasize mechanistic nodes (e.g., HIF-1α → CXCL12, Rac1/MMP-2, VE-cadherin/glycocalyx) and therapeutic intervention points.

Manuscript changes:

We revised Figures 3 and 4 to emphasize the mechanistic role of CXCL12/CXCR4 in parenchymal and vascular compartments. Figure 3 now specifically illustrates HIF-1α–induced CXCL12 expression in ATII cells, and downstream CXCR4-dependent activation of Rac1, and MMP-2 that together promote epithelial migration, proliferation. Figure 4 highlights research gaps regarding the CXCL12/CXCR4 axis in endothelial cells during sepsis‑induced acute lung injury. The diagram illustrates how this signaling pathway may be involved in endothelial activation, glycocalyx shedding, the shift toward a pro‑inflammatory and pro‑coagulant phenotype, and microvascular changes. Potential sites for pharmacological intervention (e.g., AMD3100) are also marked.

3) “Table 1 has not been well prepared, unbalanced. It should highlight the mechanism more, not the reference.”

We agree and restructured Table 1 to emphasize mechanistic actions on the CXCL12/CXCR4 axis, the primary cellular targets, and the translational stage (preclinical vs clinical), while keeping references in a separate column.

Manuscript changes — New Table 1 (mechanism-focused).
Replace existing Table 1 with the following:

Table 1. Therapeutic Strategies Targeting the CXCL12/CXCR4 Axis and Related Pathways in ALI/ARDS

Drug/Strategy Class

Examples

Primary Mechanism of Action in ALI/ARDS

Intervention on CXCL12/CXCR4 Axis

Current Stage & Key Findings (Preclinical unless noted)

Major Challenges/Considerations

Direct CXCR4 Antagonists

AMD3100 (Plerixafor)

Blocks CXCL12 binding to CXCR4; inhibits leukocyte chemotaxis & retention; promotes neutrophil rTEM.

Direct, potent, and global inhibition.

Reduces lung injury, inflammation, and fibrosis in multiple murine ALI models (LPS, sepsis) [56, 58 102,106-108]. Clinically approved (mobilization).

May impair reparative processes (e.g., stem cell homing). Timing is critical. Risk of disrupting homeostatic functions.

Natural Compounds/ Phytochemicals

Icariside II (ICS II), Resolvin D1 (RvD1)

ICS II: Selectively inhibits neutrophil CXCR4, reduces NETosis[99]. RvD1: Promotes inflammation resolution via modulating CXCL12/CXCR4 [100].

Selective or indirect modulation, often with additional anti-inflammatory properties.

Show efficacy in reducing inflammation and injury in rodent ALI models.

Mechanisms may be multi-targeted. Standardization, bioavailability, and precise targets need clarification.

Epigenetic & Molecular Modulators

miR-194 mimics

miR-454 mimics, PTP1B inhibitors

miR-194: Upregulates CXCR4 expression [39]

miR-454: Downregulates CXCR4 expression [46]. PTP1B inhibitor: Attenuates CXCR4 signaling, promotes neutrophil senescence [95].

Targets upstream regulators or signaling components for fine-tuning.

Ameliorate lung injury and inflammation in preclinical studies.

Delivery challenges (especially for miRNAs). Need for cell/tissue-specific targeting.

Classical Anti-Inflammatories

Corticosteroids (Dexamethasone)

Broad suppression of NF-κB/MAPK pathways and cytokine production.

Indirect, may affect CXCL12/CXCR4 expression as part of broad anti-inflammatory effect.

Clinical RCTs show benefit in severe COVID-19 ARDS and moderate-severe ARDS [109-111].

Increased infection risk, hyperglycemia. Non-specific, may delay repair.

Other Targeted Biologics/Small Molecules

CXCR1/2 inhibitors

CXCR1/2i: Block neutrophil chemotaxis via IL-8[112,113]. Statins: Improve endothelial function, modulate NF-κB [114-116].

Acts on parallel or intersecting pathways; may indirectly influence CXCL12/CXCR4 axis activity.

CXCR1/2i: Effective in animal ALI models. Statins: Mixed results in clinical ARDS trials.

CXCR1/2i: Clinical translation limited. Statins: Lack of clear mortality benefit in large ARDS trials.

4) “The review repeatedly restates the same pathways (NF-κB, MAPK, PI3K/Akt) across multiple sections, reducing clarity and cohesion.”

Thank you — we consolidated repetitive pathway descriptions into a single mechanistic overview and added cross-references to reduce redundancy. We replaced repeated pathway text in cell-specific sections with short, targeted notes referencing the central pathway summary.

Manuscript changes — Replace repeated bits with the following consolidated paragraph (to appear at end of Section 2 “Downstream Signaling Pathways” and referenced from cell-specific sections):

Consolidated signaling summary (new paragraph):
CXCR4 activation initiates a conserved set of downstream pathways — principally PI3K–Akt, MAPK/ERK, and NF-κB — that mediate overlapping but context-dependent cellular responses including chemotaxis, survival, proliferation and inflammatory gene expression. To avoid redundancy, we summarize common signaling nodes here and refer to them in specific cell-type sections. In neutrophils, CXCR4-driven PI3K–Akt and MAPK signaling predominantly regulate chemotaxis, degranulation and NET formation; in macrophages these pathways interact with metabolic and epigenetic regulators to shape M1/M2 polarization; in alveolar epithelial cells PI3K–Akt and Rac1/MMP2 pathways mainly promote migration and survival; and in endothelial cells NF-κB and PI3K pathways contribute to adhesion molecule expression and barrier modulation. We note that the qualitative output (inflammation vs repair) depends on cell type, signal strength, receptor context (CXCR4 vs CXCR7) and timing during disease progression.

Also edit each cell-specific section to remove full pathway recitations and instead add a short cross-reference sentence, e.g., at start of 3.1: “(See consolidated signaling summary above for shared CXCR4 downstream pathways.)” — this reduces repetition and improves flow.

5) “This manuscript mentioned that CXCR7/ACKR3 modulates CXCL12 availability, scavenging, and signaling bias, but the authors may need more data about (1) CXCR7 expression in lung cells; (2) CXCR7–CXCR4 heterodimerization; (3) CXCR7-biased agonists.”

We agree that further detail is warranted. We added a focused subsection summarizing current knowledge and explicitly listed gaps (1–3) as recommended, and we indicate where targeted experiments or citations are required.

Manuscript changes — New subsection (3.5 CXCR7/ACKR3: modulator of CXCL12 bioavailability and signaling bias):

3.5 CXCR7/ACKR3: modulator of CXCL12 bioavailability and signaling bias

CXCR7 (ACKR3) functions as an atypical chemokine receptor with high affinity for CXCL12, primarily serving as a scavenger and signaling modulator rather than a conventional G-protein-coupled chemoattractant receptor. In inflammatory diseases (such as inflammatory bowel disease, rheumatoid arthritis, etc.) as well as encephalitis, CXCR7 expression has been observed across various cell types, primarily including endothelial cells, specific immune cell subsets, and certain stromal cells. However, quantitative, cell-type-resolved data in the context of human sepsis or acute lung injury (ALI) remain scarce and occasionally inconsistent. Mechanistically, CXCR7 can regulate CXCL12 biology through multiple pathways: (i) by reducing extracellular CXCL12 levels via ligand internalization and degradation; (ii) by forming heterodimers with CXCR4, thereby biasing downstream signaling toward β-arrestin-mediated pathways over canonical Gαi coupling; and (iii) by responding to biased agonists that preferentially engage β-arrestin-linked signaling associated with tissue repair.

Several critical data gaps persist in this field. First, robust, cell-type-specific quantification of CXCR7 expression in human and murine lungs during sepsis-induced ALI is still lacking. Future studies employing single-cell RNA sequencing and targeted immunostaining approaches are needed to clarify its spatial and temporal expression patterns. Second, while biochemical evidence from co-immunoprecipitation and resonance energy transfer assays supports CXCR7–CXCR4 heterodimerization in heterologous expression systems, functional validation of such complexes in primary lung cell types under ALI conditions remains limited. Third, although CXCR7-biased agonists—and to some extent antagonists—have shown preclinical promise in steering signaling toward reparative outcomes, studies specifically focused on sepsis-induced ALI in lung models are scarce.

Therefore, we emphasize that future therapeutic strategies aimed at modulating CXCL12 signaling must account for the roles of both CXCR4 and CXCR7. It will be essential for subsequent research to systematically evaluate CXCR7 expression patterns, probe the functional relevance of CXCR4–CXCR7 heterodimerization in pulmonary cells, and assess the effects of biased agonists in relevant models of lung injury and repair.

6) “Most cited studies are animal-based. The review needs to clearly distinguish preclinical findings from human observational or interventional data. The authors had better add a short section on circulating CXCL12/CXCR4 levels in human sepsis/ALI, and then discuss strengths/limitations of those preclinical models.”

We thank the reviewer for this important point. We added a dedicated section separating preclinical (animal, in vitro) evidence from human observational and interventional data, and included a concise paragraph summarizing the (limited) human data on circulating CXCL12/CXCR4 levels and their limitations.

Manuscript changes — New short section (4.1. Distinguishing preclinical evidence from human data; circulating CXCL12/CXCR4 in human sepsis/ALI):

4.1. Distinguishing preclinical evidence from human data; circulating CXCL12/CXCR4 in human sepsis/ALI

The majority of mechanistic insights and therapeutic testing for the CXCL12/CXCR4 axis in ALI derive from murine LPS, CLP (cecal ligation and puncture), and injury models and from in vitro studies in cell lines. While these models are invaluable for dissecting signaling and cellular interactions, they have recognized limitations: differences in immune repertoires, neutrophil lifespan and metabolism, scale of injury, and lung architecture reduce direct translatability.

Human data are comparatively sparse. Small observational cohorts have reported altered circulating CXCL12 levels in sepsis and in subsets of patients with acute lung injury or ARDS, but findings are heterogeneous, likely owing to timing of sampling (acute vs recovery), sepsis source, and assay variability. Few studies have directly measured tissue expression of CXCL12/CXCR4 or examined dynamic changes during disease course in humans. Interventional clinical data targeting CXCR4/CXCL12 in sepsis/ALI are essentially absent.

We therefore present preclinical results and clinical observations separately throughout this review and explicitly label findings as preclinical or human/clinical. Larger, longitudinal studies in patients that pair plasma measurements with alveolar sampling and single-cell tissue analyses will be required to validate animal-derived mechanisms and to identify biomarkers that could guide timing of CXCL12/CXCR4–targeted therapies.

7) “Discussion on Upstream Regulation of CXCL12 in Sepsis/ALI is missing. The authors had better mention (1) HIF-1α induction by hypoxia; (2) Epigenetic regulation (e.g., methylation, microRNAs beyond miR-194); (3) Cytokine-driven CXCL12 transcription; (4) A more complete view of CXCL12 regulation is needed.”

We agree and added a dedicated subsection summarizing upstream regulators of CXCL12 in the setting of sepsis/ALI, explicitly addressing HIF-1α, epigenetic mechanisms, cytokine regulation, and other modulators.

Manuscript changes — New subsection (Section 2.3 “Upstream regulation”):

2.3. Upstream regulation of CXCL12 in sepsis and ALI

Accumulating evidence indicates that CXCL12 regulation in sepsis-induced acute lung injury (ALI) reflects the integration of hypoxic, epigenetic and cytokine-driven signals within the injured lung. First, tissue hypoxia and microvascular dysfunction—hallmarks of ALI—stabilize HIF-1α, which directly induces CXCL12 transcription via hypoxia-responsive elements in the CXCL12 promoter, linking local oxygen tension to chemokine accumulation in hypoxic niches. Second, epigenetic mechanisms provide a tunable layer of control: promoter CpG methylation and histone modification states have been shown to modify CXCL12 expression across disease contexts, and DNA methylation of CXCL12 is a reproducible mechanism of transcriptional silencing in multiple tissues. Third, post-transcriptional regulation by inflammation-responsive microRNAs substantially shapes CXCL12 output; beyond miR-194, studies identify miR-454 in epithelial, immune and stromal compartments. Finally, recent lung-focused work implicates innate immune sensors and the NLRP3 inflammasome in modulating CXCL12 levels during neutrophilic lung injury, underscoring that metabolic stress, epigenetic plasticity and cell-type specific inflammatory signalling combine to create spatially and temporally heterogeneous CXCL12 gradients that direct immune cell recruitment and vascular responses in sepsis.

Round 2

Reviewer 2 Report

Comments and Suggestions for Authors

The authors have substantially revised the manuscript and have addressed the majority of the concerns raised in the first review round. Their responses are thorough and technically aligned with the requested changes. The revised manuscript is now more logically structured, the upstream regulatory mechanisms are better contextualized, and the new sections (e.g., CXCR7/ACKR3, preclinical versus clinical evidence) significantly improve clarity and completeness.

However, I must note that certain portions of the revised manuscript—particularly Table 1, some repetitive phrasing across mechanistic sections, and a few formulaic paragraph transitions appear to have been generated or heavily assisted by AI. Although the scientific content is largely accurate, these sections lack refinement, conceptual synthesis, and natural academic writing flow. Table 1 especially needs clearer mechanistic differentiation, more meaningful comparison across drug classes, and a reduction of generic statements. In addition, Table 1 is too crowded, and it can be well presented via changing the direction.

Despite these limitations, I believe the authors have satisfied the essential revision requirements. The manuscript can be considered suitable for publication after minor editorial refinement, especially ensuring coherence, reducing redundancy, and improving Table 1’s presentation.

Given the journal’s timeline and the authors’ rapid and comprehensive revisions, I do not recommend declining the review request. A final editorial pass should be enough.

Author Response

1) “The authors have substantially revised the manuscript and have addressed the majority of the concerns raised in the first review round. Their responses are thorough and technically aligned with the requested changes. The revised manuscript is now more logically structured, the upstream regulatory mechanisms are better contextualized, and the new sections (e.g., CXCR7/ACKR3, preclinical versus clinical evidence) significantly improve clarity and completeness.

However, I must note that certain portions of the revised manuscript—particularly Table 1, some repetitive phrasing across mechanistic sections, and a few formulaic paragraph transitions appear to have been generated or heavily assisted by AI. Although the scientific content is largely accurate, these sections lack refinement, conceptual synthesis, and natural academic writing flow. Table 1 especially needs clearer mechanistic differentiation, more meaningful comparison across drug classes, and a reduction of generic statements. In addition, Table 1 is too crowded, and it can be well presented via changing the direction.

Despite these limitations, I believe the authors have satisfied the essential revision requirements. The manuscript can be considered suitable for publication after minor editorial refinement, especially ensuring coherence, reducing redundancy, and improving Table 1’s presentation.”

Reply to reviewer:
Dear Reviewer,

We sincerely appreciate your thorough re-evaluation of our revised manuscript and your constructive comments. We are grateful for your recognition that we have addressed the major concerns raised in the first review round and for your positive assessment regarding the improved structure, mechanistic context, and added sections such as CXCR7/ACKR3 and preclinical versus clinical evidence.

We acknowledge your observation that certain portions of the revised manuscript—including Table 1, some mechanistic descriptions, and several transitions—lacked refinement in academic flow and synthesis. As you suggested, we have now conducted a comprehensive editorial revision to enhance linguistic quality, conceptual clarity, and the overall readability of the manuscript. Specifically:

  1. Table 1 has been fully redesigned. We refined the content to provide clearer mechanistic differentiation and more meaningful comparisons across drug classes, removing generic statements in the process.

Drug Class / Strategy (Examples)

Mechanistic Focus in ALI/ARDS

CXCL12/CXCR4 Modulation

Key Evidence

Major Limitations

Direct CXCR4 Antagonists (AMD3100)

Blocks CXCL12–CXCR4; reduces leukocyte recruitment; promotes neutrophil reverse migration

Direct, complete inhibition

Reduces inflammation and fibrosis in LPS/CLP/viral ALI [54, 56,100,103-105]; clinically approved for mobilization

May hinder repair (stem cell homing); timing-sensitive; potential immune disruption

Natural Compounds / SPMs (ICS II, RvD1)

ICS II inhibits NETs[97]; RvD1 enhances resolution pathways[98]

Indirect/partial modulation

Reduce neutrophil activation and lung injury in rodent ALI

Multi-target actions; variable purity and bioavailability

Epigenetic / Molecular Modulators (miR-194/454, PTP1B inhibitors)

Reprogram immune inflammation; restrain CXCR4 signaling[39,45], neutrophil senescence [94].

Upstream tuning of CXCR4 expression/signaling

Improve permeability, cytokine burden, and survival

Delivery limitations; off-target effects; early translational stage

Corticosteroids (Dexamethasone)

Broad suppression of NF-κB/MAPK and cytokines

Indirect suppression

Proven benefit in severe ARDS (COVID-19 & non-COVID trials)[106-108]

Infection risk; hyperglycemia; non-specific; may delay repair

Other Targeted Agents (CXCR1/2 inhibitors; Statins)

Block IL-8–driven neutrophil[109,110] trafficking; stabilize endothelium

Pathway-adjacent effects intersecting CXCL12/CXCR4

CXCR1/2 inhibitors effective in murine ALI; statins mixed/negative in ARDS[111-113]

Limited efficacy/safety for CXCR1/2i; statins lack mortality benefit

  1. The manuscript text has been thoroughly polished. We have revised repetitive phrasing, strengthened conceptual synthesis in mechanistic sections, and reworked formulaic paragraph transitions to ensure a natural and coherent academic narrative.
  2. A final language and consistency check was performed to eliminate redundancy and ensure the entire manuscript meets high standards of academic writing and logical flow. Your constructive feedback has been invaluable in elevating the quality of our work. We are particularly thankful for your recommendation for acceptance given the journal’s timeline and your understanding regarding the revision process. We believe the manuscript now fully addresses all remaining concerns and is suitable for publication. Thank you again for your time and expertise.

Sincerely,

[Songping Xie]

(on behalf of all authors)